# Exploring and Benchmarking Planning Capabilities of Large Language Models

## Abstract

Classical and natural language planning tasks remain a difficult domain for modern large language models (LLMs). In this work, we lay the foundations for improving planning capabilities of LLMs. First, we construct a comprehensive benchmark suite encompassing both classical planning benchmarks and natural language scenarios. This suite includes algorithms to methodically generate instances of tasks with varying levels of difficulty, allowing for rigorous and systematic evaluation of LLM performance. Next, we investigate the use of many-shot in-context learning to enhance LLM planning, exploring the relationship between increased context length and improved planning performance. In addition, we demonstrate the positive impact of fine-tuning LLMs on optimal planning paths. We also probe the efficacy of chain-of-thought reasoning methods to improve LLM planning performance. Moreover, we probe the performance of the proposed methods in out-of-distribution scenarios, assessing the ability to generalize to novel and unseen planning challenges. Finally, we investigate model's failure modes and reveal insights that hold true across different benchmarks.

## 1 Introduction

Intelligent agents require the ability to plan to proactively chart a course of action to achieve their objectives. This capacity for strategic foresight is considered fundamental to intelligent behavior (Russell & Norvig, 2016). While classical search algorithms have long been the cornerstone of planning studies, machine learning techniques, particularly Monte-Carlo Tree Search (MCTS) and reinforcement learning, have emerged as useful additions, significantly expanding the capabilities of modern planning systems.

With the advent of powerful large language models (LLMs), there are new opportunities to both revisit classical planning problems, and to further explore new problems through natural language specification that reflects the ambiguity and uncertainty of real-world domains. Planning capability is important for many tasks such as game playing, meeting scheduling and trip planning. Research is already underway to leverage the commonsense knowledge of LLMs in real-world tasks (Huang et al., 2022; Singh et al., 2023; Ding et al., 2023) and to generate sensible plans (Valmeekam et al., 2023; Hao et al., 2023; Guan et al., 2024). This research has shed some light on LLMs' struggle with planning tasks. Even state-of-the-art LLMs may produce ineffective or even incorrect plans, even in straightforward scenarios. Our paper focuses on analyzing and improving the planning capability of LLM systems.

We provide a scalable benchmark suite in both PDDL and natural language to measure planning capability of LLMs. Specifically, we explore two distinct planning representations: the formal Planning Domain Definition Language (PDDL) (McDermott et al., 1998), which provides a standardized representation for classical planning problems and allows for rigorous plan validation; and natural language, which offers a more flexible and intuitive representation better reflecting real-world scenarios. For both scenarios, we provide a code for generating as many instances with a degree of difficulty of choice. We also provide a mapping method for translating PDDL benchmarks to natural language and measure the performance of the generated benchmarks. The generated planning tasks are scalable and can grow to examine and assist stronger models. Figure 1 depicts a simple sample of BlocksWorld benchmark with description both in PDDL and natural language. Each instance in BlocksWorld consists of a set of blocks, a table and a robot hand, where the goal is to move from

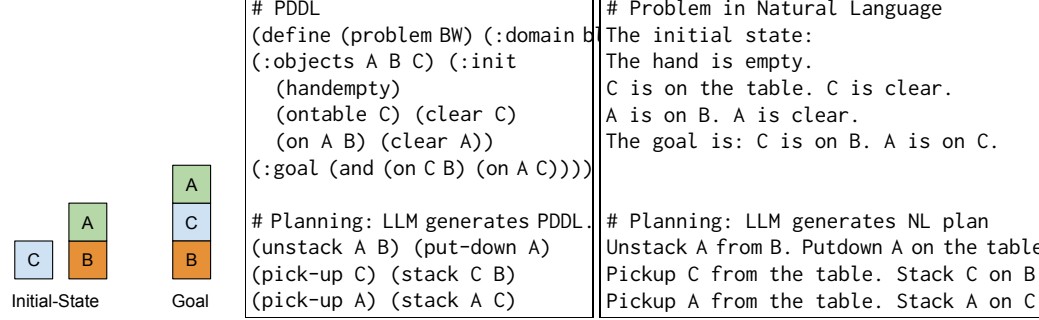

Figure 1: A simple instance of BlocksWorld planning benchmark: visual, PDDL and natural language descriptions side by side. Problem definition includes list of objects, initial and goal states. We have also included LLM's output plan in the in-context learning scenario of Figure 2.

one block configuration to another. The problem definition includes list of objects, initial and goal states. The figure also includes the LLM's output plan from the in-context learning scenario, which we extensively analyze using various LLM models and benchmarks in Figure 2.

We explore the planning capability of LLMs using both In Context Learning (ICL) through the many-shot paradigm as well and through chain-of-thought (CoT) (Wei et al., 2022) inference time techniques; we also explore fine-tuning strategies. We observe that carefully instructing the model using ICL leads to a significant boost in planning performance, which can be further improved by using a many-shot approach with long context. Moreover, CoT reasoning strategies (MCTS, Tree-of-thought, debate-as-reasoning) allow smaller models to perform closer to SoTA frontier models in the natural language task domain. Our results show that fine-tuning with the optimal plan can lead to near-perfect accuracy, even when using relatively small models.

Next, we investigate both in-domain and out-of-domain generalization and note that the proposed plans demonstrate excellent in-domain generalization: instances with similar complexity are solved with similar levels of accuracy. For ICL, prompting with easier instances leads to better performance on hard instances compared to prompting with hard instances. Fine-tuning leads to better performance for both in-domain and out-of-domain scenarios, even with a much smaller model, compared to ICL with long-context.

Lastly, we probe the failure modes of the model. We categorize the failure modes into three categories: failure to satisfy environmental constraints, failure to meet the goal and failure to generate legal actions in a given state. We note that not all of these failure modes are present across all benchmarks and methods. Moreover, as the instance complexity increases the model success rate decreases. Additionally, we pinpoint failure modes that are result of biases in training data emphasizing the importance of data curation during training.

## 1.1 RELATED WORK

Prior works investigations the planning capabilities of LLMs, have found that these models struggle to solve planning tasks (Hao et al., 2023; Valmeekam et al., 2023; 2024; Kambhampati et al., 2024), In contrast, we show that LLMs can be capable of solving such tasks and one can reach near-perfect accuracy for some scenarios, with certain methods. We run experiments on the same BlocksWorld benchmark as these prior works, and generate additional difficult cases (i.e., adding more blocks). Similar to these works, we also use PDDL verifiers to compute accuracy. See Section 3.4 for details.

Xie et al. (2024) proposed a TravelPlanner benchmark and showed GPT4-Turbo can solve some of the benchmark tasks with a success rate of $0.6\%$. The TripPlanner benchmark used in our work has two main differences: it is not an agent-based environment and rather a natural language benchmark, and it has unique answers due to carefully designed constraints.

Stechly et al. (2024) suggests that LLMs are not capable of generalizing their plans on BlocksWorld if one uses chain-of-thought (CoT) (Wei et al., 2022). In this work we show positive results in terms of generalization performance.

There is another line of work that uses a hybrid approach, meaning that they either use an external tool to solve the planning tasks (Kambhampati et al., 2024; Hirsch et al., 2024), or reformulate the problem as another task such as SMT (Hao et al., 2024) and use an external tool to solve it. Lehnert et al. (2024) use $A^*$ as search mechanism and a specific transformer architecture to achieve planning capability for that specific architecture. We differ from this line of work in that we focus on teaching LLM itself to perform the planning task.

In addition to standard few-shot prompting, we also make use of inference time techniques that construct a chain-of-thought (Wei et al., 2022), namely tree-of-thought (ToT) (Yao et al., 2023) and Monte-Carlo Tree Search (MCTS) (Hao et al., 2023) and debate-as-reasoning (Du et al., 2023) We demonstrate that these methods can considerably improve an LLMs planning capabilities such that smaller open-source models that use these approaches can outperform larger foundational models.

## 2 BENCHMARK

To evaluate the planning capabilities of LLMs, we assemble a benchmark suite that appropriately represents various classical and natural-language planning tasks. On these benchmarks, we assess LLM performance using In-Context Learning (ICL), Supervised Fine-Tuning (SFT), and chain-of-thought (CoT) methods for planning.

**PDDL (Planning Domain Definition Language):** PDDL McDermott et al. (1998) is a standardized language used in artificial intelligence for representing planning problems. PDDL provides a formal way to describe the initial state of an environment, a goals, the space of valid actions, and the state-transition properties of actions in the environment. PDDL has two main components (1) Domain: Describes the general characteristics of the planning problem, including the types of objects, actions, and predicates (conditions that can be true or false). (2) Problem: Defines a specific instance of the planning problem within the domain, including the initial state of the world and the goals to be achieved.

We select **three datasets that use PDDL**, generated additional subsets of them, and additionally map these datasets to natural language for an additional evaluation task. Additionally, we select **two native natural language datasets**, containing Trip Planning and Calendar Scheduling tasks (Zheng et al., 2024). For the PDDL-based datasets, we select BlocksWorld, Logistics, and Mini-Grid. We then translated the PDDL problem descriptions from these datasets into natural language to compare performance when using formal and informal representations.

### 2.1 PDDL BENCHMARKS

The creation of all PDDL datasets follows a three-step procedure. (1) Initially, the process involves the creation of an initial state and a goal (target state). (2) Subsequently, the initial state and goal are utilized to formulate a problem in PDDL. (3) Finally, the problem is solved using a classic planner `Fast-Downward`[1].

This procedure is repeated with increasingly difficult configurations for a select number of problems. The result of this procedure are additional datasets that comprise a set of PDDL problems and solutions of various difficulty. Importantly, this procedure enables us to create datasets with increasingly difficult problems and any number of samples, which are appropriate for assessing the ability to plan using different methods, such as in-context learning versus Supervised Fine-Tuning. Moreover, we can scale the dataset generation and create as many instances as needed for different investigations. We provide the details of our benchmark suite below.

We perform planning for BlocksWorld, Logistics and Minigrid both for PDDL and Natural Language. For the mapping to natural language, we use a slot filling technique which maps each predicate of the initialization and goal as well the action to sentences (Appendix B.2). For the verification of the plans, we use regular expressions to map the plan in Natural Language back to PDDL.

**BlocksWorld:** BlocksWorld is a standard planning problem from International Planning Conference (IPC)-2000 [2]. This domain consists of a set of blocks, a table and a robot hand, where the goal is to

---

[1] https://github.com/aibasel/downward

[2] https://github.com/potassco/pddl-instances/tree/master/ipc-2000

move from one block configuration to another. We generate a dataset for 3 to 7 blocks. As detailed in the Appendix B.1, we produced 28k unique BlocksWorld samples. From these, 25.5k were randomly selected for the training set and 2,500 for the validation set.

**Logistics:** Logistics is an AI planning problem from IPC-1998 [3] expressed in PDDL that involves arranging the delivery of packages to their destinations using trucks within cities and airplanes between cities. The aim is to optimize transportation modes under constraints such as vehicle capacities and locations, showcasing model's ability to manage multi-step logistics efficiently.

**Mini-Grid:** Mini-Grid is a task from Artificial Intelligence Planning Systems (AIPS)-1998 [4], also expressed in PDDL. We create various floorplans with rooms containing random configurations of key shapes. The goal then is for a robot to navigate from an initial position to a designated goal cell.

## 2.2 NATIVE NATURAL LANGUAGE PLANNING BENCHMARKS

**Trip Planning:** Trip Planning is a task from NaturalPlan (Zheng et al., 2024) benchmark focusing on planning a trip itinerary under given constraints. The goal of the task is to find an itinerary satisfying constraints such as the order of visiting N cities. It includes enough constraints for each instance such that there is only one solution to the task, which makes the evaluation of the predictions straightforward.

**Calendar Scheduling:** Calendar Scheduling from the NaturalPlan (Zheng et al., 2024) benchmark represents the task of scheduling a meeting of either 30 minutes or an hour among up to 7 attendees. The attendees may have a busy schedule or a light schedule with less than half of the working hours spent in meetings.

## 3 EXPERIMENTS

Previous works demonstrated that, without intervention, LLMs often struggle with even simple planning tasks (Hao et al., 2023; Valmeekam et al., 2023; 2024; Kambhampati et al., 2024). LLMs often lack the information on how to structure their plan constructively, and struggle to plan around the enumerated constraints. In this section we present our experimental results for the interventions we investigate, demonstrating that they lead to significant improvements to the LLMs planning capability. We also investigate plan generalization (i.e., the ability to generalize to unseen instances) in several scenarios.

For PDDL experiments, we measure accuracy of the generated plan with a verifier (Fox & Long). For natural language experiments we rely on either recasting the task to PDDL and verifying with a verifier, or extracting the answer and comparing it to expected results (Zheng et al., 2024). In all experiments, GPT-4 refers to GPT-4 Turbo and we may omit "Turbo" for space constraints, also if we do not mention Pro or Flash Gemini 1.5 refers to Gemini 1.5 Pro.

### 3.1 IN-CONTEXT LEARNING

For in-context learning (Brown et al., 2020), we adhere to the standard procedure, employing a prompt containing several examples for the task. Each example comprises a planning problem statement and its corresponding solution, referred to as a shot. Following the examples, the test problem is added without the corresponding solution. Subsequently, an LLM receives this prompt and is expected to generate a plan following the format and logic of the examples in the prompt. See Appendix A for examples of prompts.

#### 3.1.1 MANY-SHOT IN-CONTEXT LEARNING

We evaluate the planning capability of the model as we add more examples ("shots") into the context, inspired by the success of many-shot learning across a large number of tasks (Agarwal et al., 2024). The challenge of "in-context planning" involves understanding a specific task and problem through a limited number of examples. Additionally, it requires the models to produce a solution without

---

[3]https://github.com/potassco/pddl-instances/tree/master/ipc-1998

[4]https://github.com/AI-Planning/pddl-generators/tree/main/minigrid

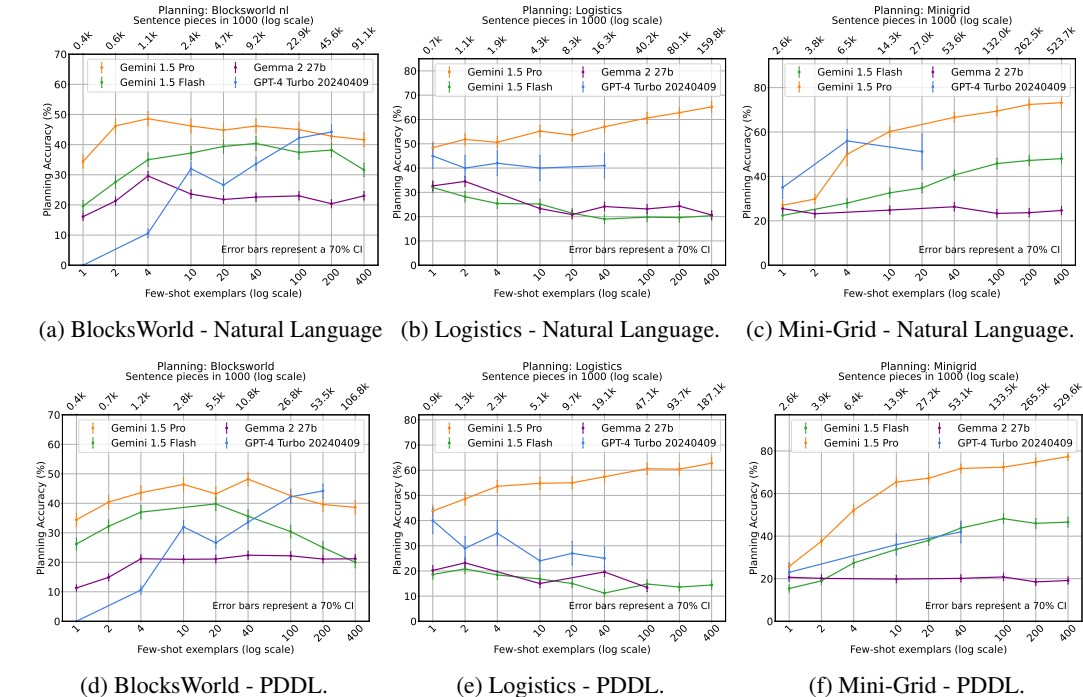

Figure 2: PDDL Planning and Natural Language Planning with few-shots for different LLMs. Natural language text are generated from formal PDDL problem definitions.

checking each planning step to confirm if a proposed move is correct. The model has to create a plan in a single inference step, keeping 'in mind' all the constraints the task imposes.

Figure 2 shows the in-context learning performance on classical planning and natural language benchmarks as we vary the number of shots. We consider Gemini 1.5 Pro (GeminiTeam et al., 2024b), GPT-4 Turbo (OpenAI et al., 2024), Gemini 1.5 Flash (GeminiTeam et al., 2024b) and Gemma 2 27b (Team et al., 2024) models. Overall, we notice that for natural language and PDDL scenarios, models have similar trends in terms of planning accuracy as we increase the number of shots. Moreover, different models are impacted differently as we provide additional number of shots, e.g., Gemini 1.5 Pro outperforms other models both in one shot scenario and as we increase the number of shots; indicating that the model not only can plan better with a fewer number of examples/shots, it can also make effective use of additional and longer context. Gemini 1.5 Flash -a smaller, faster and more efficient model than Gemini 1.5 Pro is generally outperformed by Gemini 1.5 Pro but occasionally matches GPT-4 performance.

BLOCKSWORLD: Figure 2a, 2d show the performance of Gemini 1.5 models on this benchmark as we increase the number of few-shot examples. We note that as we increase the number of shots GPT-4 performance increases while Gemini 1.5 Pro's performance saturates or degrades as we go beyond 40 shots. The 1-shot planning capability of Gemini 1.5 Pro and Gemini 1.5 Flash reaches reaches 35% and 26%, while GPT-4 performance is close to zero. Moreover the 40-shots planning capability of Gemini 1.5 Pro reaches 48% range which performs better than the best (200-shots) performance of GPT-4, which peaks at 43%.

LOGISTICS: The planning capabilities of GPT-4 and Gemini 1.5 models on the Logistics benchmark are shown in Figure 2e for PDDL and in Figure 2b for Natural Language. The 1-shot planning capability of Gemini 1.5 Pro reaches 43% for PDDL and for Natural Language 48%. Moreover for Gemini 1.5 Pro increasing the context consistently lead to better results, indicating that the model can make effective use of additional contexts. For Gemini 1.5 Flash and GPT-4, the performance drops slight for PDDL and Natural Language.

MINI-GRID: Figure 2f and Figure 2c show the performance of GPT-4 and Gemini models as we increase the number of few-shot examples for PDDL and Natural Language, respectively. The Gemini models perform comparably for both PDDL and Natural Language, although GPT-4 appears

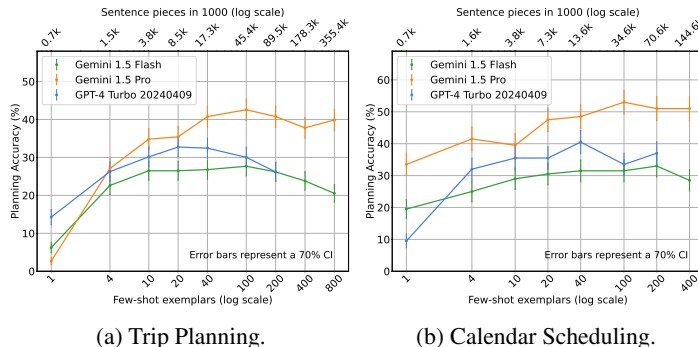

(a) Trip Planning.    (b) Calendar Scheduling.

Figure 3: Natural Language Planning with few-shots using native natural language datasets.

to perform slightly better with Natural Language. Increasing the number of few-shot examples leads to better performance for all models. With 400 shots, Gemini 1.5 Pro reaches 77% accuracy.

TRIP PLANNING AND CALENDAR SCHEDULING: Figure 3a shows the performance on the Trip Planning and Calendar Scheduling natural language tasks as we increase the number of few-shot examples, respectively. We observe that, in both benchmarks, for both GPT-4 and Gemini 1.5 Flash the performance first increases with the number of shots and after a certain point, having more shots leads to worse model performance. However, for Gemini 1.5 Pro performance improves as the number of shot increases. Therefore, Gemini 1.5 Pro seems to be making more efficient use of additional shots compared to the other two. On the other hand GPT-4 performs better in the 1-shot scenario compared to the other two models for Trip Planning, while Gemini 1.5 Pro has a higher accuracy in the 1-shot setting.

Overall, we observe that the trend of accuracy vs number of shots depends both on the model and on the benchmark.

### 3.1.2 EFFECT OF INFERENCE TIME TECHNIQUES

In addition to standard ICL, we consider inference time ICL methods that are based on constructing a chain-of-thought (Wei et al., 2022): Tree-of-Thought (ToT) (Yao et al., 2023), Monte Carlo Tree Search (MCTS) (Hao et al., 2023), and Debate-as-reasoning (Du et al., 2023). In Figure 4 we provide experimental evidence that methods such as Debate-as-reasoning, MCTS, and ToT can augment Gemma2 27B (considerably smaller than Gemini 1.5 and GPT-4 models), to be competitive with GPT-4, Gemini 1.5 Pro and Gemini 1.5 Flash at smaller few-shot context lengths. Additional details and parameters for these search methods are included in Appendix B.3. These results demonstrate how inference-time reasoning procedures using smaller open-source models can perform competitively with larger foundational models in the natural language planning domain.

However, the ability of these methods to scale to few-shot examples seems to be less significant than larger foundational models. Specifically, for the Travel Scheduling task, the larger models all

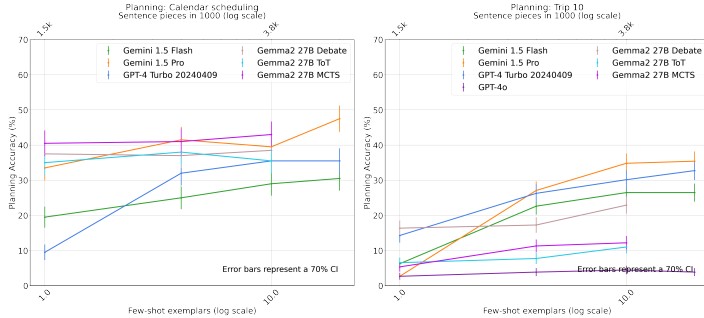

Figure 4: Calendar Scheduling (left) and Travel Scheduling (right) tasks with reasoning procedures (ToT, MCTS, and Debate). We use Gemma2 27B to perform these procedures, and compare with GPT-4 and Gemini 1.5 models.

outperform the CoT methods after 4 few-shot examples. It is unclear if this scaling trend is a result of the underlying model (Gemma2 27B) or the method itself.

Most interesting to note is the performance of debate-as-reasoning, which, despite not being an explicit search or planning strategy, works comparatively to MCTS in both tasks. This indicates that, for some planning tasks, even unstructured CoT style inference methods are competitive with explicit search-based CoT methods, and larger foundational models with ICL. This may indicate that, for natural language planning tasks, allowing the model to construct a CoT and consider multiple solutions may be more valuable then the specific method behind constructing the CoT.

## 3.2 SUPERVISED FINE TUNING (SFT)

Supervised fine tuning (SFT) (Ouyang et al., 2022) has proved to be an effective method for teaching LLMs various capabilities. In this section, we investigate the effect of Supervised Fine Tuning (SFT) with optimal plan on planning capability of LLMs. We use the Fast-Downward classical planner to generate the optimal plan. We specifically ran experiments on Gemini 1.0 S (GeminiTeam et al., 2024a) and investigate planning performance for two different benchmarks with different levels of difficulty, namely, we look into 5 scenarios: BlocksWorld with 3-7 blocks, 8-9 blocks or 8-20 blocks, and Logistics with 1-2 packets, or with 3-5 packets. The data size and splits are documented in Appendix B.4.

The results are shown in Table 1. We observe that SFT leads to high accuracy for some instances of both datasets and outperforms many-shot ICL. The performance appears to drop as the planning problem becomes more difficult.

Table 1: Impact of SFT on accuracy, for BlocksWorld: instance of 3-7, 8-9 and 8-20 blocks, and for Logistics: instances of 1-2 and 3-5 packets.

| Model | Gemini 1.0 S |
|---|---|
| BlocksWorld(3-7) | 96.26 |
| BlocksWorld(8-9) | 92.6 |
| BlocksWorld(8-20) | 67.00 |
| Logistics(1-2) | 99.8 |
| Logistics(3-5) | 63.4 |

Table 2: Plan generalization analysis for instances of BlocksWorld of different number of blocks in SFT scenarios, for Gemini 1.0 S. Accuracy is in %. Accuracy is measured by a verifier.

| Finetune data | Eval data | Accuracy |
|---|---|---|
| BW(3-7) | BW(3-7) | 96.26 |
| BW(3-7) | BW(8-20) | 34.20 |
| BW(8-20) | BW(3-7) | 98.27 |
| BW(8-20) | BW(8-20) | 67.00 |

## 3.3 PLAN GENERALIZATION

For any LLM application, the question of how well the method generalizes to out-of-training-distribution (OOD) inputs is always present. Here, we investigate how LLM planning generalizes. Plan generalization has three main categories: (1) Generalize to unseen instances of the same environment (2) Generalize to renaming of actions and objects (3) Generalize to unseen plan environments. We focus on the first category which sits at the core of the desired capability.

For these generalization experiments, we consider BlocksWorld and Logistics benchmarks with various difficulty levels (as described in Section 3.2). We look into performance of both SFT and ICL approaches. Tables 2 shows generalization performance for BlocksWorld in SFT setting for BlocksWorld 3-7, 8-20 split and Table 3 considers generalization between BlocksWorld 3-7 and 8-9 blocks split for both SFT and ICL with different models and number of shots. Table 4 depicts plan generalization for Logistics benchmark for splits of 1-2 and 3-5 packets.

Our analysis reveals several key findings: (1) Superiority of SFT: SFT consistently outperforms ICL across both benchmarks, even when utilizing a smaller model for SFT. This suggests that SFT's explicit training process, focused on the specific task, leads to more effective learning and better generalization. (2) In most ICL scenarios, training the model on easier instances first results in improved performance on harder examples, e.g., see Table 3, rows 1-4 and 6. (3) Limitations of hard example training: Contrary to some expectations, training the model exclusively on hard examples does not always translate to better performance on easier ones (for example, see Table 3 rows 2, 4-6).

This suggests that a balanced approach, incorporating both easy and hard examples, might be optimal for achieving well-rounded performance.

Table 3: OOD accuracy for BlocksWorld splits 3-7 and 8-9 blocks.

| Model Name | Train data | |
|---|---|---|
| | 3-7 | 8-9 |
| | Eval data | |
| | 3-7 / 8-9 | 3-7 / 8-9 |
| Gemini 1.5 Flash (1 Shot) | 26.4 / 13.7 | 32.9 / 12.3 |
| Gemini 1.5 Flash (70 Shot) | 35.7 / 23.3 | 27.6 / 11.0 |
| Gemini 1.5 Pro (1 Shot) | 40.0 / 25.6 | 39.0 / 20.3 |
| Gemini 1.5 Pro (70 Shot) | 46.3 / 36.3 | 38.3 / 18.4 |
| Gemini 1.0 S (1 Shot) | 3.68 / 0.331 | 2.99 / 1.35 |
| Gemini 1.0 S (70 Shot) | 12.4 / 2.33 | 3.99 / 1.68 |
| Gemini 1.0 S (SFT) | 96.3 / 81.6 | 96.0 / 92.6 |

Table 4: OOD accuracy for Logistics tasks splits of 1-2, 3-5 packets.

| Model Name | Train data | |
|---|---|---|
| | 1-2 | 3-5 |
| | Eval data | |
| | 1-2 / 3-5 | 1-2 / 3-5 |
| Gemini 1.5 Flash (1 Shot) | 18.3 / 1.35 | 26.7 / 1.00 |
| Gemini 1.5 Flash (30 Shot) | 12.7 / 1.67 | 19.7 / 1.33 |
| Gemini 1.5 Pro (1 Shot) | 35.3 / 9.03 | 57.6 / 7.01 |
| Gemini 1.5 Pro (30 Shot) | 56.4 / 11.3 | 62.7 / 8.04 |
| Gemini 1.0 S (1 Shot) | 7.0 / 0.0 | 5.33 / 0.0 |
| Gemini 1.0 S (30 Shot) | 9.99 / 0.336 | 8.00 / 0.662 |
| Gemini 1.0 S (SFT) | 99.8 / 10.8 | 98.0 / 63.4 |

## 3.4 COMPARISON WITH PLANBENCH

Valmeekam et al. (2023) proposed a benchmark for planning that maps domain definitions to instructions and problem statements into natural language using zero-shot and one-shot techniques. We utilize their dataset on BlocksWorld, as the problems are comparable. Unlike their approach, which limits problems to configurations of 3, 4, and 5 blocks using only zero-shot and one-shot prompting, our work extends this using for ICL up to 7 blocks and by employing many-shot prompting.

Table 5 compares results using the natural language prompts from Valmeekam et al. (2023)(their dataset is referred to as *Val-BW*) and, novel to this work, presents results on PDDL for their datasets using both 1-shot and 2-shot techniques.

We utilize the natural language prompts from (Valmeekam et al., 2023) test them on Gemini 1.5 Pro. We observe that GPT-4 performs better with these prompts. For our dataset, no such difference is observed. Manual inspection reveals that especially 1-shot prompts need to be crafted carefully while

Table 5: Accuracy (in %) comparing the state of the art for different datasets and systems. Val-BW denotes BlocksWorld dataset as open-source by Valmeekam et al. (2023).

| Dataset | LLM | Shots | Type | No. of Blocks | | |
|---|---|---|---|---|---|---|
| | | | | 3 | 4 | 5 |
| Val-BW | GPT-4 Turbo | 1 | NL | 49.0 | 32.4 | 23.2 |
| Val-BW | Gemini 1.5 Pro | 1 | NL | 30.0 | 18.4 | 14.2 |
| Val-BW | Gemini 1.5 Pro | 1 | PDDL | 60.0 | 36.4 | 23.6 |
| Val-BW | Gemini 1.5 Pro | 2 | PDDL | 68.0 | 46.2 | 30.9 |
| Our-BW | Gemini 1.5 Pro | 1 | NL | 66.0 | 38.5 | 32.4 |
| Our-BW | Gemini 1.5 Pro | 2 | NL | 66.0 | 58.5 | 53.9 |
| Our-BW | Gemini 1.5 Pro | 1 | PDDL | 100 | 44.6 | 32.0 |
| Our-BW | Gemini 1.5 Pro | 2 | PDDL | 100 | 44.6 | 50.9 |

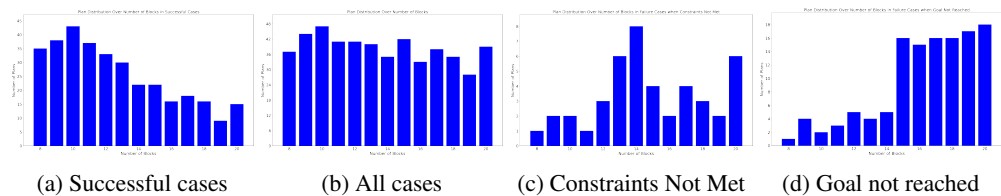

(a) Successful cases     (b) All cases     (c) Constraints Not Met     (d) Goal not reached

Figure 5: In-domain failure analysis: Distribution of number of blocks in successful and failed cases and different failure reasons for BlocksWorld 8-20 blocks. As the number of blocks increases the number of successful cases decreases.

few-shot or many-shot prompts are more robust. For instance, results improve when the prompts are more specific about the output format, changing from '*My plan is as follows*' to '*Your plan as plain text without formatting*', which enhances results for Gemini 1.5 Pro. Further, our prompts do not include explanations of the actions and we do not use color coding for the blocks but rather keep the names (e.g., blue block vs block b3).

## 4 INVESTIGATING FAILURE CASES

In this section, we analyze the outcome of various approaches to improving LLM planning performance, and dive into a more detailed categorization of failure modes.

The LLM planning failure modes we observed can be classified into the following three categories:

**(1)** Constraint Violation: The model fails to satisfy one of the explicitly declared environment constraints. i.e., it ignores one of the conditions required to be able to do an action. Even if this happen only for one instance of an action and not for all instances of the same action. For example model wants to put block b1 on block b2 while block b2 is not empty.

**(2)** Failure to meet the goal: The model makes a plan that meets the environment constraints but does not reach the goal state at the end of the trajectory.

**(3)** Out-of-vocabulary actions: The model proposes actions which are not in the environment's action space.

**PDDL benchmarks** Here we examine the failure modes for both the Blocksworld and Logistics domains and SFT, ICL setting. Due to space constraints we report Blocksworld SFT settings here and refer the reader to Appendix C for the rest of plots and analysis. First we look at the effect of distribution of number of blocks in the planning instance. We compare Figure 5a that includes successful eval cases of BlocksWorld 8-20 blocks to, to all its eval data in Figure 5b respectively. We note that the model failure cases increases as the number of blocks in the problem increases. We also separate the reasons of failure in BlocksWorld 8-20 instances as seen in Figure 5c and Figure 5d respectively.Note that failure mode (3) does not happen for BlocksWorld benchmark in SFT or ICL settings and for Logistics in ICL setting, but for Logistics benchmark in SFT settings all three categories are present (see Appendix C for details).

Next, We study the distribution of number of blocks in out-of-domain cases focusing on BlocksWorld 3-7 to 8-20 blocks generalization. Comparing the successful cases in Figure 6a to all the eval samples in Figure 5b, we observe that the generalization performance of the model that is trained on 3-7 blocks drops as the number of eval blocks increases from 8 to 20.

To probe the models even further, we look into the category where the generated plan fails to satisfy one of the environment constraints. We study the step at which the plan fails to meet the constraint. Considering Figure 6b and comparing it to Figure 6c, the model seems to have trouble generalizing from the beginning, having the majority of its failures concentrated on earlier steps. This trend stays true for Logistics benchmark as well and is due to the fact that models go deep before they go wide when they want to produce the plan output. In other terms, this gives the intuition that the model seems to get overwhelmed when encountering difficult unseen examples, leading to failure from the beginning by not satisfying the constraints.

In addition, looking at the granular level of which actions that cause the failure in Figure 6b, we note that the model has learned a correlation between the step number and the action it chooses, leading

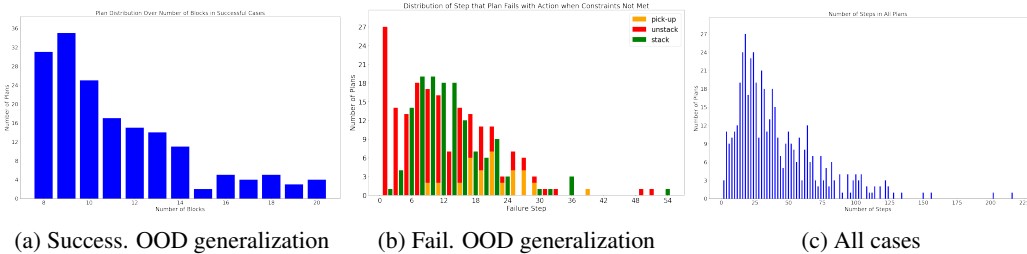

(a) Success. OOD generalization     (b) Fail. OOD generalization     (c) All cases

Figure 6: BlocksWorld 3-7 to 8-20 OOD scenario (a)Success per number of blocks (b) investigation of failures in constraint violation per plan step, color coded by action where the constraint violation happens (c) distribution of all cases per plan steps.

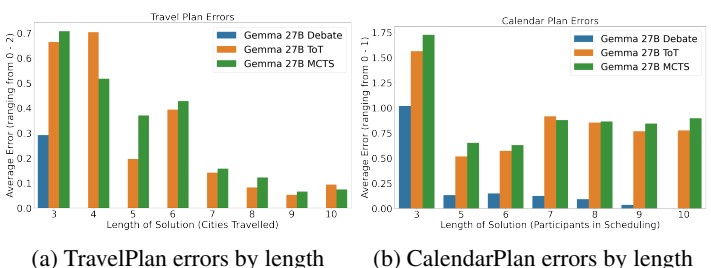

(a) TravelPlan errors by length     (b) CalendarPlan errors by length

Figure 7: Error Analysis for various CoT reasoning strategies in the TravelPlan and CalendarPlan natural language tasks

to having stack and un-stack not having any mutual failure step. This is also observed in failure modes of BlocksWorld 3-7 both for ICL and SFT scenarios (see Figure 11 in Appendix C). Further investigation of training data confirms this observation.

**Natural Language benchmarks** Since TravelPlan and CalendarPlan benchmarks require plans that do not involve sequential execution of actions, the only failure mode we consider is failure mode (2) - failure to reach a goal state. All errors discussed in this subsection are in terms of whether or not the method under study succeeded in reaching the goal. Figure 7 shows the average error by CoT method as task complexity increases in the TravelPlan and CalendarPlan benchmarks. Figure 12 in Appendix C histograms of how many examples belong to each task length. Both domains have a higher distribution of "smaller" or shorter tasks, which impacts the distribution of error in Figure 7. It is interesting to note how the error is disproportionate between debate-as-reasoning and MCTS, ToTs, aligning with our findings in Section 3. As discussed in Section 3, methods such as MCTS and ToT seem to perform inconsistently at less-structured planning tasks. We can see this in how they perform relative to debate, as well as each other. Specifically, for the TravelPlan task, they fail at inconsistent rates relative to one another.

## 5 CONCLUSION

We examined planning capabilities of LLMs through benchmark development, generalization assessment and analysis of failure modes. Our observations for ICL setting has implications for the future development and training of LLMs, potentially informing strategies to enhance their capacity to process and leverage extended contextual information. It also points to the opportunity for followup work investigating ways to further improve the inference time reasoning procedures, and their scaling properties. Our investigation of plan generalization reveals three key findings: superiority of SFT, curriculum learning effectiveness and limitations of hard example training; suggesting that a balanced approach, incorporating both easy and hard examples, might be optimal for achieving well-rounded performance. Our analysis of failure modes point to future work on dataset design and reasoning procedures to directly address the discussed failure modes. These enhancements can unlock new levels of versatility and robustness in LLM-based planning systems, paving the way for their broader adoption in real-world applications.

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

# Appendix

## A    PROMPTS

**Bellow is the 1-shot prompt for the BlocksWorld task.**

```
Please solve the problem:
(define (problem BW-rand-4)
(:domain blocksworld-4ops)
(:objects b4 b1 b3 b2)
(:init
(on b3 b1)
(on b1 b4)
(clear b3)
(handempty)
(ontable b2)
(ontable b4)
(clear b2)
)
(:goal (and
(on b2 b4)
(on b3 b1)
))
)

Your plan as plain text without formatting:
(unstack b3 b1)
(put-down b3)
(unstack b1 b4)
(put-down b1)
(pick-up b2)
(stack b2 b4)
(pick-up b3)
(stack b3 b1)
done.

Please solve the problem:
(define (problem BW-rand-6)
(:domain blocksworld-4ops)
(:objects b5 b1 b4 b2 b3 b6)
(:init
(on b4 b1)
(handempty)
(ontable b6)
(on b2 b4)
(clear b3)
(ontable b5)
(on b3 b2)
(clear b6)
(on b1 b5)
)
(:goal (and
(on b4 b2)
(on b1 b4)
(on b5 b1)
(on b3 b5)
))
)

Your plan as plain text without formatting:
```

**Bellow is the 1-shot prompt for the Logistics task.**

```
Please solve the problem:
(define (problem logistics-c4-s2-p3-a4)
(:domain logistics-strips)
(:objects
a0 a1 a2 a3
c0 c1 c2 c3
t0 t1 t2 t3
l0-0 l0-1 l1-0 l1-1 l2-0 l2-1 l3-0 l3-1
p0 p1 p2
)
(:init
  (AIRPLANE a0) (AIRPLANE a1)(AIRPLANE a2)(AIRPLANE a3)
  (CITY c0)(CITY c1)(CITY c2)(CITY c3)
  (TRUCK t0)(TRUCK t1)(TRUCK t2)(TRUCK t3)
  (LOCATION l0-0)(in-city   l0-0 c0)
  (LOCATION l0-1)(in-city   l0-1 c0)
  (LOCATION l1-0)(in-city   l1-0 c1)
  (LOCATION l1-1)(in-city   l1-1 c1)
  (LOCATION l2-0)(in-city   l2-0 c2)
  (LOCATION l2-1)(in-city   l2-1 c2)
  (LOCATION l3-0)(in-city   l3-0 c3)
  (LOCATION l3-1)(in-city   l3-1 c3)
  (AIRPORT l0-0)(AIRPORT l1-0)(AIRPORT l2-0)(AIRPORT l3-0)
  (OBJ p0)(OBJ p1)(OBJ p2)
  (at t0 l0-0)(at t1 l1-1)(at t2 l2-0)(at t3 l3-0)
  (at p0 l1-1)(at p1 l0-1)(at p2 l0-0)
  (at a0 l1-0)
  (at a1 l1-0)
  (at a2 l2-0)
  (at a3 l3-0)
)
(:goal
  (and
    (at p0 l2-0)
    (at p1 l2-0)
    (at p2 l1-1)
  )
)
)

Your plan as plain text without formatting:
(load-truck p0 t1 l1-1)
(drive-truck t1 l1-1 l1-0 c1)
(unload-truck p0 t1 l1-0)
(load-airplane p0 a1 l1-0)
(fly-airplane a1 l1-0 l2-0)
(unload-airplane p0 a1 l2-0)
(drive-truck t0 l0-0 l0-1 c0)
(load-truck p1 t0 l0-1)
(drive-truck t0 l0-1 l0-0 c0)
(unload-truck p1 t0 l0-0)
(fly-airplane a3 l3-0 l0-0)
(load-airplane p2 a3 l0-0)
(fly-airplane a3 l0-0 l1-0)
(unload-airplane p2 a3 l1-0)
(load-truck p2 t1 l1-0)
(drive-truck t1 l1-0 l1-1 c1)
(unload-truck p2 t1 l1-1)
(fly-airplane a1 l2-0 l0-0)
(load-airplane p1 a1 l0-0)
(fly-airplane a1 l0-0 l2-0)
(unload-airplane p1 a1 l2-0)
```

```
756    done.
757
758
759    Please solve the problem:
760    (define (problem logistics-c2-s2-p3-a2)
761    (:domain logistics-strips)
       (:objects
762    a0 a1
763    c0 c1
764    t0 t1
765    l0-0 l0-1 l1-0 l1-1
       p0 p1 p2
766    )
767    (:init
768       (AIRPLANE a0)(AIRPLANE a1)
769       (CITY c0)(CITY c1)
770       (TRUCK t0) (TRUCK t1)
771       (LOCATION l0-0)(in-city  l0-0 c0)
       (LOCATION l0-1)(in-city  l0-1 c0)
772       (LOCATION l1-0)(in-city  l1-0 c1)
773       (LOCATION l1-1)(in-city  l1-1 c1)
774       (AIRPORT l0-0) (AIRPORT l1-0)
775       (OBJ p0)(OBJ p1)(OBJ p2)
       (at t0 l0-1)(at t1 l1-0)
776       (at p0 l0-1)(at p1 l1-0)(at p2 l1-1)
777       (at a0 l0-0)(at a1 l0-0)
778    )
779    (:goal
780       (and
         (at p0 l0-1)
781         (at p1 l1-0)
782         (at p2 l0-0)
783       )
784    )
785    )
786
787    Your plan as plain text without formatting:
788
789
790    Bellow is the 1-shot prompt for the Mini-Grid task.
791
792    Please solve the problem:
793    (define (problem grid_2Vroom2)
794       (:domain grid)
       (:objects
795         p0 p1 p2 p3 p4 p5 p6 p7 p8
796         shape0
         key0
797       )
798       (:init
799         ; Object types
800         (place p0) (place p1) (place p2) (place p3) (place p4) (place p5) (
             place p6) (place p7) (place p8)
801         (shape shape0)
802         (key key0)
803         ; Open/locked cells
804         (open p0) (open p1) (open p2) (open p3) (open p5) (open p6) (open p7)
             (open p8)
805         (locked p4)
806         ; Connected cells
807         (conn p0 p1)
808         (conn p0 p2)
809         (conn p1 p0)
         (conn p1 p3)
```

```
810        ( conn p2 p0 )
811        ( conn p2 p3 )
812        ( conn p2 p4 )
813        ( conn p3 p2 )
814        ( conn p3 p1 )
815        ( conn p4 p2 )
816        ( conn p4 p5 )
817        ( conn p5 p4 )
818        ( conn p5 p6 )
819        ( conn p5 p7 )
820        ( conn p6 p5 )
821        ( conn p6 p8 )
822        ( conn p7 p5 )
823        ( conn p7 p8 )
           ( conn p8 p7 )
           ( conn p8 p6 )
           ; Lock and key shapes
           ( lock−shape p4 shape0 )
           ( key−shape key0 shape0 )
           ; Key placement
           ( at key0 p0 )
           ; Robot placement
           ( at−robot p3 )
           ( arm−empty )
        )
      ( : goal ( at−robot p7 ) )
    )

Your plan as plain text without formatting:
( move p3 p2 )
( move p2 p0 )
( pickup p0 key0 )
( move p0 p2 )
( unlock p2 p4 key0 shape0 )
( move p2 p4 )
( move p4 p5 )
( move p5 p7 )
done .

Please solve the problem :
( define ( problem grid_3Vroom3 )
   ( : domain grid )
   ( : objects
     p0 p1 p2 p3 p4 p5 p6 p7 p8 p9 p10 p11 p12 p13 p14 p15 p16 p17 p18 p19
         p20 p21 p22 p23 p24 p25 p26 p27 p28
     shape0
     key0
   )
   ( : init
     ; Object types
     ( place p0 ) ( place p1 ) ( place p2 ) ( place p3 ) ( place p4 ) ( place p5 ) (
         place p6 ) ( place p7 ) ( place p8 ) ( place p9 ) ( place p10 ) ( place p11
         ) ( place p12 ) ( place p13 ) ( place p14 ) ( place p15 ) ( place p16 ) (
         place p17 ) ( place p18 ) ( place p19 ) ( place p20 ) ( place p21 ) ( place
         p22 ) ( place p23 ) ( place p24 ) ( place p25 ) ( place p26 ) ( place p27 )
         ( place p28 )
     ( shape shape0 )
     ( key key0 )
     ; Open/ locked cells
     ( open p0 ) ( open p1 ) ( open p2 ) ( open p3 ) ( open p4 ) ( open p5 ) ( open p6 )
         ( open p7 ) ( open p8 ) ( open p10 ) ( open p11 ) ( open p12 ) ( open p13 )
         ( open p14 ) ( open p15 ) ( open p16 ) ( open p17 ) ( open p18 ) ( open p20 )
         ( open p21 ) ( open p22 ) ( open p23 ) ( open p24 ) ( open p25 ) ( open p26
         ) ( open p27 ) ( open p28 )
     ( locked p9 ) ( locked p19 )
```

```
; Connected cells
( conn p0 p1 )
( conn p0 p3 )
( conn p1 p0 )
( conn p1 p2 )
( conn p1 p4 )
( conn p2 p1 )
( conn p2 p5 )
( conn p3 p0 )
( conn p3 p4 )
( conn p3 p6 )
( conn p4 p3 )
( conn p4 p1 )
( conn p4 p5 )
( conn p4 p7 )
( conn p5 p4 )
( conn p5 p2 )
( conn p5 p8 )
( conn p6 p3 )
( conn p6 p7 )
( conn p6 p9 )
( conn p7 p6 )
( conn p7 p4 )
( conn p7 p8 )
( conn p8 p7 )
( conn p8 p5 )
( conn p9 p6 )
( conn p9 p10 )
( conn p10 p9 )
( conn p10 p11 )
( conn p10 p13 )
( conn p11 p10 )
( conn p11 p12 )
( conn p11 p14 )
( conn p12 p11 )
( conn p12 p15 )
( conn p13 p10 )
( conn p13 p14 )
( conn p13 p16 )
( conn p14 p13 )
( conn p14 p11 )
( conn p14 p15 )
( conn p14 p17 )
( conn p15 p14 )
( conn p15 p12 )
( conn p15 p18 )
( conn p16 p13 )
( conn p16 p17 )
( conn p17 p16 )
( conn p17 p14 )
( conn p17 p18 )
( conn p18 p17 )
( conn p18 p15 )
( conn p18 p19 )
( conn p19 p18 )
( conn p19 p22 )
( conn p20 p21 )
( conn p20 p23 )
( conn p21 p20 )
( conn p21 p22 )
( conn p21 p24 )
( conn p22 p21 )
( conn p22 p19 )
( conn p22 p25 )
( conn p23 p20 )
```

```
        (conn p23 p24)
        (conn p23 p26)
        (conn p24 p23)
        (conn p24 p21)
        (conn p24 p25)
        (conn p24 p27)
        (conn p25 p24)
        (conn p25 p22)
        (conn p25 p28)
        (conn p26 p23)
        (conn p26 p27)
        (conn p27 p26)
        (conn p27 p24)
        (conn p27 p28)
        (conn p28 p27)
        (conn p28 p25)
        ; Lock and key shapes
        (lock-shape p9 shape0)
        (lock-shape p19 shape0)
        (key-shape key0 shape0)
        ; Key placement
        (at key0 p12)
        ; Robot placement
        (at-robot p16)
        (arm-empty)
    )
    (:goal (at-robot p4))
)
```

Your plan as plain text without formatting:

**Bellow is the 1-shot prompt for the Trip Planning task.**

Please solve the problem:
You plan to visit 6 European cities for 13 days in total. You only take
    direct flights to commute between cities. You want to spend 3 days in
     Dublin. You would like to meet your friends at Dublin between day 7
    and day 9 to tour together. You would like to visit Madrid for 2 days
    . You plan to visit relatives in Madrid between day 2 and day 3. You
    plan to stay in Oslo for 3 days. You would like to visit London for 2
     days. You want to spend 3 days in Vilnius. You plan to stay in
    Berlin for 5 days. You are going to attend a wedding in Berlin
    between day 3 and day 7.

Here are the cities that have direct flights:
London and Madrid, Oslo and Vilnius, Berlin and Vilnius, Madrid and Oslo,
     Madrid and Dublin, London and Oslo, Madrid and Berlin, Berlin and
    Oslo, Dublin and Oslo, London and Dublin, London and Berlin, Berlin
    and Dublin.

Find a trip plan of visiting the cities for 13 days by taking direct
    flights to commute between them.

Here is the trip plan for visiting the 6 European cities for 13 days:

**Day 1-2:** Arriving in London and visit London for 2 days.
**Day 2:** Fly from London to Madrid.
**Day 2-3:** Visit Madrid for 2 days.
**Day 3:** Fly from Madrid to Berlin.
**Day 3-7:** Visit Berlin for 5 days.
**Day 7:** Fly from Berlin to Dublin.
**Day 7-9:** Visit Dublin for 3 days.
**Day 9:** Fly from Dublin to Oslo.

**Day 9−11:** Visit Oslo for 3 days.
**Day 11:** Fly from Oslo to Vilnius.
**Day 11−13:** Visit Vilnius for 3 days.
done.

Please solve the problem:
You plan to visit 6 European cities for 17 days in total. You only take direct flights to commute between cities. You want to spend 4 days in Manchester. You plan to stay in Florence for 5 days. You want to spend 3 days in Geneva. You are going to attend a wedding in Geneva between day 1 and day 3. You want to spend 3 days in Seville. During day 7 and day 9, you have to attend a conference in Seville. You would like to visit Prague for 2 days. You plan to stay in Valencia for 5 days. From day 3 to day 7, there is a annual show you want to attend in Valencia.

Here are the cities that have direct flights:
Manchester and Prague, Seville and Manchester, Geneva and Manchester, Valencia and Seville, Geneva and Valencia, Valencia and Prague, Prague and Florence, Geneva and Prague.

Find a trip plan of visiting the cities for 17 days by taking direct flights to commute between them.

**Bellow is the 1-shot prompt for the Calendar Scheduling task.**

Please solve the problem:
You need to schedule a meeting for Samuel, Evelyn, Ruth and Amanda for half an hour between the work hours of 9:00 to 17:00 on Monday.

Here are the existing schedules for everyone during the day:
Samuel is free the entire day.
Evelyn has meetings on Monday during 9:00 to 10:00, 11:00 to 12:00, 12:30 to 13:00, 15:30 to 16:00;
Ruth has meetings on Monday during 9:30 to 11:00, 11:30 to 12:30, 13:00 to 13:30, 14:00 to 14:30, 15:00 to 16:00, 16:30 to 17:00;
Amanda has meetings on Monday during 10:00 to 10:30, 11:00 to 12:30, 13:00 to 13:30, 14:00 to 15:00, 15:30 to 16:00;

Amanda can not meet on Monday before 16:00. Find a time that works for everyone's schedule and constraints.

Here is the proposed time: Monday, 16:00 − 16:30
done.

Please solve the problem:
You need to schedule a meeting for Walter, Jacob, Jennifer and Joan for one hour between the work hours of 9:00 to 17:00 on Monday.

Here are the existing schedules for everyone during the day:
Walter is busy on Monday during 9:30 to 10:00, 13:00 to 13:30;
Jacob has meetings on Monday during 11:00 to 11:30, 13:00 to 13:30;
Jennifer is busy on Monday during 9:30 to 10:30, 11:30 to 12:00, 12:30 to 15:00;
Joan has blocked their calendar on Monday during 9:30 to 10:00, 10:30 to 11:30, 12:00 to 12:30, 13:00 to 14:00, 14:30 to 15:30;

Find a time that works for everyone's schedule and constraints.

## B  EXPERIMENTAL DETAILS

### B.1  DATASET CREATION

In these experiments BlocksWorld dataset for 3 to 7 blocks consists of 40000 samples.

In the creation of the BlocksWorld dataset as outlined in Algorithm 1, the key parameters include the maximum number of blocks $num\_blocks$ and the quantity of examples $n$ to be generated for each block count. Here, the maximum number of blocks is a number greater than 3. As we use uniform sampling, this results in a linear increase in the number of more complex examples. However, it's important to note that as the number of blocks increases, the simpler combinations are exhausted since all possible combinations might be included. The methods *CreateStacks* generates random stacks of blocks, iteratively sampling from the available blocks to determine stack heights until all blocks are utilized. The method *CreatePro* denotes a simple method to translate the block configuration into PDDL which is python reimplementation of functionality in 4ops-Blockworld code}[5].

---

**Algorithm 1** Create BlocksWorld Dataset

---

**function** CREATEDATASETBW($num\_blocks$, $n$)
    $dataset \leftarrow []$                                          ▷ Initialize an empty list
    **for** $problem\_id \leftarrow 1$ **to** $n$ **do**
        $b \leftarrow$ RANDOMUNIFORM(3, $num\_blocks$)
        $initStacks \leftarrow$ CREATESTACKS($b$)
        $goalStacks \leftarrow$ CREATESTACKS($b$)
        **if** $initStacks == goalStacks$ **then**
            **continue**                                  ▷ Skip equal stacks.
        **end if**
        $problem \leftarrow$ CREATEPRO($initStacks, goalStacks$)
        $plan \leftarrow$ FASTDOWNWARD($problem, domain$)
        $dataset \leftarrow dataset + [(problem, plan)]$
    **end for**
    **return** $dataset$
**end function**

---

Algorithm 1, we generate 28k unique samples. From these, we randomly select 25500 of the for training set and 2500 for validation set. This procedure yields a problem distribution as shown in Figure 8.

### B.2  MAPPINGS PDDL TO NATURAL LANGUAGE

Here we present the templates to map PDDL problems to Natural Language. Details are shown in Table 6.

### B.3  SEARCH PROCEDURE PARAMETERS

.

The two search procedures deployed and compared alongside ICL and SFT methods, (ToT) (Yao et al., 2023) and monte-carlo tree search (MCTS) (Hao et al., 2023), were implemented as specified in their original papers. The only deviations are listed below.

The biggest deviation from the reference papers are the LLM's prompts, which had to be edited to make the search procedures more aligned with the planning task.

Additionally, for the MCTS procedure, the action log-probs were weighted by a factor of 1.5. All other weights specified in the Reasoning as Planning MCTS procedure are the same (state log-probs, UCT, and exploration lambda factor are all 1.0).

The same weights are used to compute the value of the nodes in the tree-of-thought search procedure.

---

[5]https://github.com/AI-Planning/pddl-generators/tree/main/blocksworld/4ops

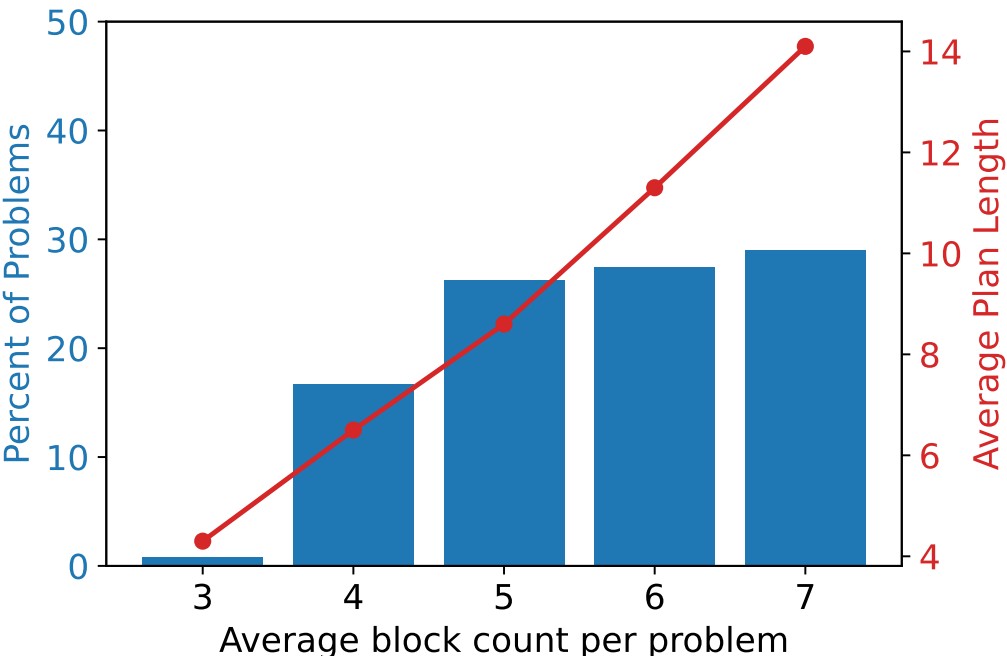

Figure 8: Distribution with number of blocks and average plan length.

| Term (with arguments) | Mapping to Natural Language |
| --- | --- |
| AIRPLANE $object_2$ | $object_2$ is an AIRPLANE. |
| CITY $object_2$ | $object_2$ is a CITY. |
| TRUCK $object_2$ | $object_2$ is a TRUCK. |
| at $object_2$ $object_3$ | $object_2$ is at $object_3$. |
| in-city $object_2$ $object_3$ | $object_2$ is in the city $object_3$. |
| drive-truck $param_2$ $param_3$ $param_4$ $param_5$ | Drive truck $param_2$ from $param_3$ to $param_4$ in $param_5$. |
| load-truck $param_2$ $param_3$ $param_4$ | Load $param_2$ into truck $param_3$ at $param_4$. |
| unload-truck $param_2$ $param_3$ $param_4$ | Unload $param_2$ from truck $param_3$ in $param_4$. |
| fly-airplane $param_2$ $param_3$ $param_4$ | Fly airplane $param_2$ from $param_3$ to $param_4$. |
| load-airplane $param_2$ $param_3$ $param_4$ | Load $param_2$ into airplane $param_3$ at $param_4$. |
| unload-airplane $param_2$ $param_3$ $param_4$ | Unload $param_2$ from airplane $param_3$ at $param_4$. |
| on $object_2$ $object_3$ | $object_2$ is on $object_3$. |
| handempty | The hand is empty. |
| ontable $object_2$ | $object_2$ is on the table. |
| clear $object_2$ | $object_2$ is clear. |
| unstack $param_2$ $param_3$ | Unstack $param_2$ from $param_3$. |
| put-down $param_2$ | Put down $param_2$. |
| pick-up $param_2$ | Pick up $param_2$. |
| stack $param_2$ $param_3$ | Stack $param_2$ on $param_3$. |
| conn $object_2$ $object_3$ | $object_2$ and $object_3$ are connected. |
| lock-shape $object_2$ $object_3$ | The lock $object_2$ is $object_3$ shaped. |
| key-shape $object_2$ $object_3$ | The key $object_2$ is $object_3$ shaped. |
| arm-empty | The arm is empty. |
| open $object_2$ | $object_2$ is OPEN. |
| move $param_2$ $param_3$ | Move from $param_2$ to $param_3$. |
| pickup $param_2$ $param_3$ | Pickup $param_2$ at $param_3$. |
| unlock $param_2$ $param_3$ $param_4$ $param_5$ | Unlock $param_2$ at $param_3$ using $param_4$, which has $param_5$. |
| pickup-and-loose $param_2$ $param_3$ | At $param_2$, pick up $param_3$ and lose $param_2$. |
| at-robot $object_2$ | Robot is at $object_2$. |

Table 6: Semantic mappings used in the system, showing terms and their arguments.

| Hyperparameter | Value | Description |
|---|---|---|
| LLM Model | Gemini 1.0M | The language model used for text generation. |
| LLM Temperature | 1.0 | Controls the randomness of LLM outputs (higher values = more variance). |
| LLM Num Samples | 1 | The number of different outputs generated by the LLM for each input. |
| Max Depth | 5 | The maximum number of steps in the search tree. |
| Max Branching Factor | 3 | The maximum number of actions to consider at each node. |
| Num Simulations | 3 | The number of times to simulate the game from each node. |

Table 7: Monte Carlo Tree Search (MCTS) Hyperparameters

| Hyperparameter | Value | Description |
|---|---|---|
| LLM Model | Gemini 1.0M | The language model used for text generation. |
| LLM Temperature | 1.0 | Controls the randomness of LLM outputs (higher values = more variance). |
| LLM Num Samples | 1 | The number of different outputs generated by the LLM for each input. |
| Max Depth | 5 | The maximum number of steps in the thought process. |
| Max Branching Factor | 3 | The maximum number of alternative thoughts to explore at each step. |
| Num Simulations | 3 | The number of rollouts for each thought to simulate. |

Table 8: Tree of Thought (ToT) Hyperparameters

| Prompt Name | Prompt Content |
|---|---|
| MCTS_STATE_PROMPT | `Given the provided state and action, estimate the next state. Your state should look similar to preceding state, enclosed in [STATE] blocks. Do not repeat or reiterate information from the preceding states / actions. [STATE CONTEXT] {state} [END STATE CONTEXT] [STATE]` |
| MCTS_ACTION_PROMPT | `[CONTEXT] {state} [END CONTEXT] Given the preceding task, and action, what action should be taken next? Only take a SINGLE STEP at a time. Any composite actions will be penalized. [ACTION]` |

Table 9: MCTS Agent Prompts

### B.4 FINETUNE EXPERIMENTS

For the fine tuning of the Gemini 1.0 S, we use learning rate of 0.0001 with drop out rate of 0.1. We train the model for 5k step and choose the checkpoint with highest accuracy on the validation set. We then run the verifier on the inference results of that checkpoint and report the results.

## C ERROR ANALYSIS: ADDITIONAL PLOTS

As mentioned in Section 4, for Logistics SFT experiments the three categories of the error are all present, for example, in the Id setting for 3-5 packets, number of correct instances are 317/500 and the distribution of failure modes are 57/500, 125/500, 1/500 for categories (1), (2), (3) respectively. and in the OOD setting of 1-2 packet to 3-5 packet case, number of correct instances are 54/500 and the distribution of failure modes are 180/500, 237/500, 29/500 for categories (1), (2), (3) respectively.

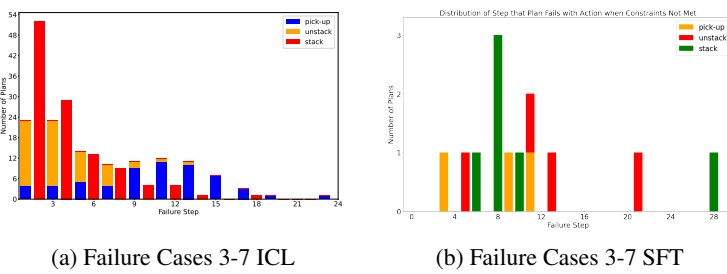

(a) Failure Cases 3-7 ICL  (b) Failure Cases 3-7 SFT

Figure 9: Side by side portray of Failure cases where constraints are not met for BlocksWorld 3-7 blocks cases in ICL and SFT scenarios per step number color coded by action name.

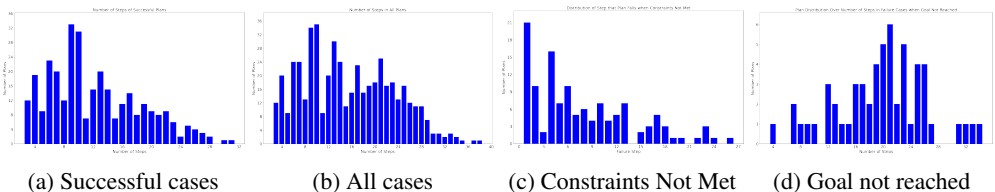

(a) Successful cases  (b) All cases  (c) Constraints Not Met  (d) Goal not reached

Figure 10: In-domain failure analysis: Distribution of number of blocks in successful and failed cases and different failure reasons for Logistics 3-5 packets. As the number of blocks increases the number of successful cases decreases.

| Prompt Name | Prompt Content |
|---|---|
| MCTS_STATE_PROMPT | `Given the provided state and action, estimate the next state. Your state should look similar to preceding state, enclosed in [STATE] blocks. Do not repeat or reiterate information from the preceding states / actions. [STATE CONTEXT] {state} [END STATE CONTEXT] [STATE]` |
| MCTS_ACTION_PROMPT | `[CONTEXT] {state} [END CONTEXT] Given the preceding task, and action, what action should be taken next? Only take a SINGLE STEP at a time. Any composite actions will be penalized. [ACTION]` |

Table 10: Tree-of-Thought Prompts

Table 11: CalendarPlan Performance with search procedures (ToT, MCTS) per number of few-shot examples provided to the procedure. We observe that for contexts fitting within Gemini 1.0M, it competes with significantly more powerful models. Without these methods, the model fails outright.

| N | G1.0M ToT | G1.0M MCTS | G1.5 Flash | GPT4 Turbo |
|---|---|---|---|---|
| 1 | 29 | 28 | 39 | 19 |
| 4 | 33 | 39 | 50 | 64 |
| 10 | 31 | 36 | 58 | 71 |

| Dataset | Train Size | Test Size |
|---|---|---|
| BW(3-7) | 28,386 | 500 |
| BW(8-9) | 3,995 | 500 |
| BW(8-20) | 4,160 | 500 |
| Logistics(1-2) | 13,483 | 500 |
| Logistics(3-5) | 13,483 | 500 |

Table 12: Details of the dataset size

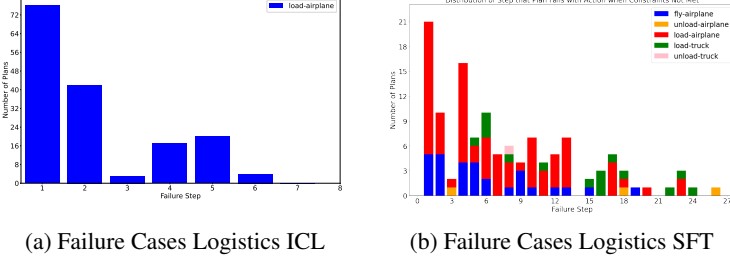

(a) Failure Cases Logistics ICL      (b) Failure Cases Logistics SFT

Figure 11: Side by side portray of Failure cases where constraints are not met for Logistics 3-5 packets cases in ICL and SFT scenarios per step number color coded by action name.

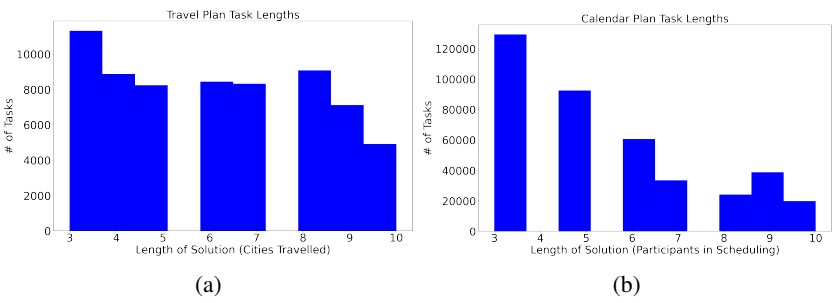

(a)            (b)

Figure 12: Histogram of # of tasks by task-length in the Travel Plan and Calendar Plan natural language tasks.

