# OpenReview forum: "Exploring and Benchmarking  Planning Capabilities of  Large Language Models"
_ICLR.cc/2025/Conference — Submitted to ICLR 2025_

### Official Review · Reviewer_4bWU · 2024-11-01

**Soundness:** 2
**Presentation:** 3
**Contribution:** 1
**Rating:** 3
**Confidence:** 5

**Summary:**

This paper runs various experiments on 3 PDDL datasets (Blocksworld, Logistics, Mini-Grid) and 2 natural language datasets (Trip Planning, Calendar Scheduling) to investigate (1) the effect of many-shot prompting on all 5 datasets (2) the effect of inference-time prompting techniques on 2 datasets (Trip Planning, Calendar Scheduling) (3) the effect of SFT using ID and OOD on 2 datasets (Blocksworld, Logistics) (4) the performance differences on PlanBench's Blocksworld/Logistics dataset versus this paper's and (5) failure case analysis.

**Strengths:**

$Originality:$ This paper runs large-scale experiments on environments explored by the LLM planning community using new models.

$Quality:$ This paper has plenty of results that substantiate past findings.

$Clarity:$ The paper is written very clearly.

$Significance:$ Some of the experiments may suggest that newer models deviate from past findings (see questions).

**Weaknesses:**

The main issue with this paper is that it has very little novelty to contribute to the LLM planning community since it reproduces many results and echoes insights that have been communicated in other papers. There is a contrast in narrative to existing work that claims LLMs cannot plan using base models with no inference time techniques [1, 2]; however, plenty of work exists that individually investigate and demonstrate that prompting with examples boosts performance [3, 4], inference-time prompting techniques boosts performance [5, 6, 7, 8], finetuning boosts performance [9, 10, 11], and do similar failure analyses [2, Figure 5].

The conclusion ends with a quick note on 3 key findings regarding plan generalization: (1) the superiority of SFT (2) curriculum learning effectiveness and (3) limitations of hard example training. These points have already been brought up in [12] that uses SFT to tackle plan generalization but uses a better approach of process supervision [13].

In addition, while there are 5 datasets total experimented on in this paper, the experiments focus mainly on Blocksworld and Logistics which has already been focused on in [2], with TravelPlan and CalendarPlan only being used for many-shot and inference-time experiments and MiniGrid only being used for the many-shot experiment. It is unclear why these 5 datasets are introduced as part of a benchmark suite but only subsets are experimented on.

Overall, this paper reads primarily as a summary of well-investigated trends in the LLM planning community. To strengthen this paper, it would be useful to propose an approach atop the experiment findings to add novelty to the paper and further support the insights (for example, the balanced approach incorporating easy and hard examples in the conclusion). Since it would require non-trivial rewriting to improve this paper, I have suggested a rating of reject.

[1] On the planning abilities of large language models-a critical investigation (Valmeekam et. al 2023)

[2] Planbench: An extensible benchmark for evaluating large language models on planning and reasoning about change (Valmeekam et. al 2023)

[3] Language Models are Few-Shot Learners (Brown et. al 2020)

[4] Many-shot in-context learning (Agarwal et. al 2024)

[5] ReAct: Synergizing Reasoning and Acting in Language Models (Yao et. al 2022)

[6] Tree of Thoughts: Deliberate Problem Solving with Large Language Models (Yao et. al 2023)

[7] Graph of Thoughts: Solving Elaborate Problems with Large Language Models (Besta et. al 2023)

[8] Language Agent Tree Search Unifies Reasoning Acting and Planning in Language Models (Zhou et. al 2023)

[9] FireAct: Toward Language Agent Fine-tuning (Chen et. al 2023)

[10] Plansformer: Generating Symbolic Plans using Transformers (Pallagani et. al 2022)

[11] LLMs Can't Plan, But Can Help Planning in LLM-Modulo Frameworks (Kambhampati et. al 2024)

[12] Beyond A*: Better Planning with Transformers via Search Dynamics Bootstrapping (Lehnert et. al 2024)

[13] Let's Verify Step by Step (Lightman et. al 2023)

**Questions:**

- What amount of overlap exists between the examples in the many-shot setting and the test examples? For example, in Blocksworld, could the many-shot examples encompass the test example (e.g. solving the same 3-block configuration in a setting with 4 blocks where the 4th block is untouched)?
- Gemini 1.5-Pro improving in performance with more shots compared to Flash and gpt-4 turbo is interesting. There is existing work [1] that shows that LLMs struggle with long contexts; how would you justify Pro appearing to be an exception to this?
- Why is ReAct and PDDL experiments missing from the inference time techniques?
- The SFT experiments are surprising to me, especially the plan generalization experiments (Table 2) finetuning on less blocks and evaluating on more blocks and vice versa. My intuition tells me that these experiments would have lower performance because of length generalization issues with LLMs [2]. Can you share the average horizon lengths for these instances and a breakdown of performance as horizon length increases?
- Have you considered running SFT on input-process-output rather than input-output to boost your OOD generalization as other works have investigated [3-4]
- Cost breakdown of experiments?

[1] Lost in the Middle: How Language Models Use Long Contexts (Liu et. al 2023)

[2] Exploring Length Generalization in Large Language Models (Anil et. al 2022)

[3] Beyond A*: Better Planning with Transformers via Search Dynamics Bootstrapping (Lehnert et. al 2024)

[4] Let's Verify Step by Step (Lightman et. al 2023)

---

### Official Review · Reviewer_MTEa · 2024-11-01

**Soundness:** 2
**Presentation:** 2
**Contribution:** 2
**Rating:** 3
**Confidence:** 5

**Summary:**

This paper presents datasets for large language model (LLM) reasoning tasks from three classical planning domains. The paper explores how LLM performs in those classical planning domains for in-context learning and fine-tuning settings. The paper also provides a comparison with a similar benchmark set, PlanBench, and the paper investigates failure cases.

**Strengths:**

In terms of originality, there are similar benchmarks such as PlanBench available in the literature.  The authors argue that the paper also considers many-shot in-context learning settings, as shown in Figure 2.  The number of few-shot exemplars ranges from 1 to 400.
Figure 4 shows results from three inference-time ICL methods: ToT, MCTS, and Debate-as-reasoning in non-PDDL-based domains.
Lastly, this paper also shows the experiment results from the fine-tuning setting.

**Weaknesses:**

On the coverage of classical planning domains.
This paper argues that the proposed work lays the foundation for improving the planning capabilities of LLMs by presenting three classical planning domains. From a classical planning viewpoint, those domains were more than 20 years old and they don't have many features in PDDL and only have almost the minimal subset of the language features. Namely, it is almost STRIPS with a negative precondition. If we view the direction of translating PDDL to NL, the work should consider the coverage of the translation of the language.

On the quality of translations.
The paper mentioned that the translation is done by regular expressions. LLMs are supposed to be trained in human-written texts and natural languages. However, the translations are done by hand-crafted regular expressions that translate formal PDDL descriptions. It is doubtful how such a dataset would enhance the reasoning capability of LLMs.
If we consider the LLM as a symbolic reasoning engine like a planner such as fast-downward, the scale of problems is so tiny that it is doubtful how the proposed methods will be useful.

**Questions:**

Question 1. Figure 2.
We see that the accuracy degrades when more examples are given in certain cases.
What conclusion can be drawn from Figure 2? The number of examples can be considered as a hyperparameter?
If so, how does that hyperparameter apply to unseen problems?

The orange curve shows similar performance in both translated problems and PDDL problems. There are a few variations, but overall the performance/trends look similar. What is the implication of these results? Should we provide PDDL as is?

How many problem instances were evaluated per each domain? How the initial state and goals were determined for the experiment? Does the figure include instances with varying numbers of objects?


Question 2. Figure 4.
What are the results from PDDL domains?

Question 3. Fine-tuning
The experiment was done on 5 problem instances. How the initial states and the goals are determined?
How the object names are given to the problem? For generalization, what parts of the problem are generalized?

Question 4. Figure 6.
Can you explain the out-of-distribution? What is the original distribution and why it is called out of distribution?

Question 5. Overall conclusion.
Does this study imply the negative reporting on using LLM for solving classical planning problems?
The overall results show that LLMs struggle to solve tiny problems and it is not clear how to resolve the issues.

---

### Official Review · Reviewer_bRrD · 2024-11-03

**Soundness:** 2
**Presentation:** 2
**Contribution:** 1
**Rating:** 1
**Confidence:** 5

**Summary:**

The paper provides a PDDL and NL planning benchmark, evaluates ICL and SFT with LLMs on this benchmark dataset, and probes for failure models and plan generalization abilities.

**Strengths:**

1. A new NL planning benchmark dataset.

**Weaknesses:**

1. Did not find the paper to have novel results (please refer to questions)
2. No details on how the plans are being generated using FD (which heuristic is being used to generate optimal plans?)
3. The NL translated prompt does not include PDDL domain, making me question how any LLM is supposed to know all actions, preconditions, and effects for a given domain?
4. Not a complete coverage of literature on plan generation. Especially, authors in [1] introduce thought of search which steers LLMs to solve 100% planning problems. [2] already provided similar conclusions on SFT using even smaller parameter models and also talked about multiple plan generalization issues.

[1] Katz, M., Kokel, H., Srinivas, K., & Sohrabi, S. (2024). Thought of Search: Planning with Language Models Through The Lens of Efficiency. In The First Workshop on System-2 Reasoning at Scale, NeurIPS'24.

[2] Pallagani, V., Muppasani, B., Murugesan, K., Rossi, F., Srivastava, B., Horesh, L., ... & Loreggia, A. (2023). Understanding the capabilities of large language models for automated planning. arXiv preprint arXiv:2305.16151.

**Questions:**

1. What is the overall purpose of using LLMs for solving planning problems? For example, to generate 400-shots to append to a prompt, assuming the 400-shots are coming from a symbolic planner, to hopefully get a correct plan out of LLM seems pointless. A survey on LLMs and planning has already pointed out multiple issues with using LLMs for plan generation.
2. What is the heuristic used to generate an optimal plan using FastDownward?
3. How distinct are the n-shots provided in ICL different to the actual question posed in the prompt? Is there any overlap in goals?
4. Why is the domain model not translated to PDDL? This is a major flaw and does not equate to planning because the authors have omitted the actions, their preconditions, and effects.
5. How does the paper differ from the related work mentioned in weaknesses?

---

### Official Review · Reviewer_63qK · 2024-11-04

**Soundness:** 1
**Presentation:** 2
**Contribution:** 1
**Rating:** 1
**Confidence:** 5

**Summary:**

This paper benchmarks the planning capabilities of Large Language Models on the PDDL and travel planning domains, and also discusses the impact of fine-tuning LLMs on optimal planning solutions. The presented experiments include results with Gemma 2 27b, Gemini 1.5 Pro, Gemini 1.5 Flash, and GPT-4-turbo, and the authors primarily analyze the planning performance of these models when varied with the number of few-shot examples in the prompt.

**Strengths:**

1) The paper presents a consolidated set of results on two recently popular natural language planning benchmarks, along with additionally generated problems in PDDL planning domains.
2) This is the first work that systematically analyzes LLM planning capabilities with fine-tuning on optimal solutions, and if fine-tuning is effective with respect to generalizing across different problem categories under a domain (for example, training on PDDL blocksworld problems with 3-7 blocks and testing with PDDL problems of 8-9 blocks, and so on).

**Weaknesses:**

1) Novelty - One of the biggest contributions that the work claims is of the planning dataset. However, the paper only combines existing PDDL planning dataset (PlanBench) and travel planning domains, with the addition of some newly generated test problems. This seems to be more like packaging the two domains together than presenting a new dataset in a paper. Can the authors provide more details to counter this argument - such as - what % of problems are new in this dataset that are not present in the two existing datasets?
2) Unjustified evaluations for improved performance - It seems that the paper aims to establish that LLMs perform exceptionally well in planning domains, but at the cost of prompt engineering (line 73 - authors say that ICL methods show boost in performance with 'carefully instructing the model') and cherry-picking evaluation data (line 89 - authors say that 'data curation during training' can be helpful to avoid biases thereby helping LLMs). Why do we have to give all the benefit of the doubt to the LLMs at the expense of human effort during prompting and data processing? It is worthwhile noting the challenges that LLMs face (using failure mode analysis) without such human efforts, as they can be constructively useful for further improving their planning capabilities.
3) Fine-tuning results - The authors show near-perfect accuracy for SFT results in Tables 3 and 4, but only in the cases where the train and the eval data consisted of the similar problem cases. This is a classic over-fitting example, and thus, does not lead to any conclusions regarding the so-called 'generalization' abilities of LLMs.

**Questions:**

1) What measures are specifically used for categorizing in-distribution and out-of-distribution samples for planning problems? Is this distribution measured only with respect to, say number of blocks in blocksworld domain?
2) Line 104 - what do the authors mean by 'agent-based environment' for the TripPlanning benchmark? Why were the results on that benchmark not shown in this work? Why is it a problem to have 'unique answers' to each planning problem - doesn't that mean that the authors rely on a set of problems that can have multiple solutions among which LLM can be attributed success if it gets any one by one way or the other?
3) What are the final takeaways from each set of the results? What are the possible reason/s or intuition/s that smaller models are able to outperform larger models? There seems to be no discussion on these points except the stated results. It will be helpful to further elaborate.

---

### Meta-Review · Area_Chair_UKL4 · 2024-12-06

**Metareview:**

This paper benchmarks the planning capabilities of LLMs on the PDDL and travel planning and also explores the impact of fine-tuning LLMs on optimal planning solutions. All reviews are negative and the main concern is the novelty as all reviewers pointed out. The paper only combines the existing PDDL planning dataset (PlanBench) and travel planning domains.

As the authors did not provide a rebuttal, it is a clear rejection.

**Additional Comments On Reviewer Discussion:**

The authors did not provide a rebuttal.

---

### Decision · Program_Chairs · 2025-01-22

Reject